# Structures of G-protein coupled receptor HCAR3 in complex with selective agonists reveal the basis for ligand recognition and selectivity

Fang Ye[1,2☸], Zhiyi Zhang[2,3☸], Binghao Zhang[2☸], Xinyu Li[4☸], Jiaxi Deng[1], Qian Miao[1], Peiruo Ning[2], Yunlin Chi[2], Geng Chen[2], Zhangsong Wu[2], Qian Wang[2], Lezhi Xu[2], Ningjie Gong[2], Bangning Cheng[4], Zhigang Ma[2]*, Chungen Qian[5]*, Lizhe Zhu[4]*, Xin Pan[6]*, Yang Du [2]*

1 Guangxi Key Laboratory of Special Biomedicine, School of Medicine, Guangxi University, Nanning, China, 2 Kobilka Institute of Innovative Drug Discovery, The Second Affiliated Hospital, Shenzhen Futian Biomedical Innovation R&D Center, School of Medicine, Chinese University of Hong Kong, Shenzhen, Guangdong, China, 3 The Huanan Affiliated Hospital of Shenzhen University, Shenzhen University, Shenzhen, Guangdong, China, 4 Warshel Institute for Computational Biology, School of Medicine, the Chinese University of Hong Kong, Shenzhen, Guangdong, China, 5 Department of Reagent Research and Development, Shenzhen YHLO Biotech Co., Ltd., Shenzhen, Guangdong, China, 6 Department of Cardiology, Central Laboratory, The Affiliated Hospital of Yangzhou University, Yangzhou University, Yangzhou, Jiangsu, China

☸ These authors contributed equally to this work.
* qmc89@163.com (ZM); chungen_qian@hust.edu.cn (CQ); zhulizhe@cuhk.edu.cn (LZ); xinpanphd@163.com (XP); yangdu@cuhk.edu.cn (YD)

## Abstract

The hydroxycarboxylic acid receptors (HCAR2 and HCAR3), also known as prototypical metabolite-sensing receptors, are key targets for treating dyslipidemia and metabolic disorders. While HCAR2 activation, but not HCAR3 activation, is associated with side effects of cutaneous flushing, the structural features and ligand preferences of HCAR3 remain less understood. Here, we used Sf9 cells to express HCAR3-Gi and HCAR2-Gi complexes, and present cryo-EM structures of HCAR3-Gi complexes with agonists compound 6O (3.31 Å), D-phenyllactic acid (3.05 Å), IBC293 (3.26 Å), and acifran (3.18Å), as well as HCAR2-Gi complex with agonist acifran (2.72 Å). Our findings reveal the mechanism behind 6O's highest affinity to HCAR3, attributed to its full occupation of both R1 and R2 regions of the orthosteric binding pocket. Moreover, combined with cAMP assay in HEK-293 cells, we have elucidated that the ligand selectivity between HCAR3 and HCAR2 depended on π–π interaction with F107[3.32] (L107[3.32] in HCAR2) and ligand-binding pocket size difference, facilitated by key residues difference V/L83[2.60], Y/N86[2.63], and S/W91[23.48]. Collectively, these structural insights lay the groundwork for developing HCAR3-specific drugs, potentially avoiding HCAR2-induced adverse effects.

**Data availability statement:** All relevant data are within the paper and its Supporting information files. The 3D cryo-EM density maps of the compound 6O, PLA, IBC293, and acifran-HCAR3-Gi1-scFv16 complexes have been deposited in the Electron Microscopy Data Bank database under accession codes EMD-61570, EMD-61571, EMD-61572, and EMD-61573, respectively. The atomic coordinates for the atomic models of the compound 6O, PLA, IBC293, and acifran-HCAR3-Gi1-scFv16 complexes generated in this study have been deposited in the Protein Data Bank database under accession codes 9JKS, 9JKT, 9JKV, and 9JKX, respectively. The 3D cryo-EM density map of the acifran-HCAR2-Gi1-scFv16 complex has been deposited in the Electron Microscopy Data Bank database under accession codes EMD-61574. The atomic coordinates for the atomic models of the acifran-HCAR2-Gi1-scFv16 complex generated in this study has been deposited in the Protein Data Bank database under accession codes 9JKY.

**Funding:** This work was supported by the National Natural Science Foundation of China (grant number: 32271263 and 82574379 (Y.D.), the Natural Science Foundation of Guangxi (grant number: 2025GXNSFBA069457 (F.Y.), and Shenzhen Sci. & Tech Innovation Bureau JCYJ 20220818103009018 (Y.D.) and JCYJ 20240813113521028 (Y.D.), the Shenzhen-Hong Kong Cooperation Zone for Technology and Innovation HZQB-KCZYB-2020056 (Y.D.), and the key grant from the Second Affiliated Hospital at the CUHK-Shenzhen (Y.D. and Z.M.). F.Y. is supported by grants from the Fund Project of Guangxi University (project code ZX01080033724008). The funders had no role in study design, data collection and analysis, decision to publish, or preparation of the manuscript.

**Competing interests:** The authors have declared that no competing interests exist.

**Abbreviations:** cAMP, cyclic adenosine monophosphate; CHS, cholesteryl hemisuccinate; cryo-EM, cryo-electron microscopy; CTF, contrast transfer function; DDM, dodecylmaltoside; DNGαi1, dominant-negative Gαi1; GDN, glycol diosgenin; GPCR, G-protein-coupled receptor;HCAR2, hydroxycarboxylic acid receptor 2; HCAR3, hydroxycarboxylic acid receptor 3; IBC293,

## Introduction

Hydroxycarboxylic acid receptors (HCAR) consist of three subtypes: HCAR1, HCAR2, and HCAR3, all of which belong to class A G-protein-coupled receptor (GPCR) family [1,2]. HCARs play crucial roles in maintaining homeostasis and regulating fundamental biological processes by binding to Gi protein, thereby inhibiting adenylyl cyclase activity and decreasing cytosol cAMP levels [3,4]. As a key regulator in metabolism, HCAR3 is widely expressed in several cell types, including adipocytes, immune cells, and intestinal epithelial cells [5,6]. In general, activation of HCAR3, by its endogenous ligand 3-hydroxyoctanoic acid, has been shown to increase high-density lipoprotein and cholesterol levels, concurrently reducing formation of macrophage foam cells to alleviate atherosclerosis [7–9]. Moreover, HCAR3 can regulate glucose metabolism and β-oxidation of fatty acid, thus is being assessed as a potential target for treating diabetes and insulin resistance. In addition to metabolism regulation, the expression level of HCAR3 is closely associated with colorectal cancer and breast cancer, hence, HCAR3 is recognized as a potential therapeutic target for these two types of cancer [10,11].

HCAR2 shares the highest sequence similarity with HCAR3 in HCAR family, and is considered to be a molecular target for regulating dyslipidemia, activated by the endogenous ligand 3-hydroxybutyric acid [12,13]. Both HCAR3 and HCAR2 have similar functions in lipid metabolism and inflammation, through facilitating the phosphorylation of ERK1/ERK2, which is a crucial step in lipolysis inhibition. However, activation of HCAR2 in Langerhans cells and keratinocytes triggers vasodilatory prostaglandins release, resulting in cutaneous flushing [14,15]. Compared to HCAR2, activation of HCAR3 can effectively control lipid levels while avoiding the side effect of flushing, making it a more promising therapeutic target for dyslipidemia treatment [16,17]. However, the structural mechanism of ligand preference between HCAR3 and HCAR2 remains unclear.

Acifran is a clinically-used drug that primarily targets both HCAR2 ($EC_{50} = 10^{-5.7}$M) and HCAR3 ($EC_{50} = 10^{-4.7}$M), to treat hypertriglyceridemia, type 2 diabetes, and metabolic syndrome [18,19]. The higher affinity of acifran to HCAR2 than to HCAR3 underscores the need to understand the structural basis for this difference, providing insights for HCAR3-selective drug development. Based on the chemical structure of acifran, two HCAR3 selective full-agonists, IBC293 and compound 6O (6O), have been developed [16,17]. IBC293 and 6O have at least 100-fold higher affinity to HCAR3 than to HCAR2. Another natural HCAR3 selective compound, D-phenyllactic acid (PLA), is found in fermented foods and produced by human gut microbiome lactic acid bacteria, which has a function of promoting human monocyte migration to potentially facilitate anti-inflammatory and immunomodulatory effects [20]. The discovery of these chemicals provides valuable tools to further explore the pharmacological and therapeutic potential of HCAR3 in lipid disorders treatment, while avoiding the side effects caused by HCAR2 activation. Elucidating the binding mode and pharmacological properties of these HCAR3 agonists can benefit development of high-affinity, low-side-effect drugs. Recently, several cryo-EM structures of HCAR3

1-propan-2-ylbenzotriazole-5-carboxylic acid;
LMNG, lauryl maltose neopentyl glycol; MD,
molecular dynamics; PLA, D-phenyllactic
acid; RMSD, root mean-square deviation;
Sf9, spodoptera frugiperda 9; TM, transmembrane; TMD, transmembrane domain;WT, wild
type;6O, 3-[bis(thiophen-3-ylmethyl)amino]-
1H-pyrazole-5-carboxylic acid.

bound to ligands 3-hydroxyoctanoic acid, niacin, compound 5c [21], acifran [22], as
well as cryo-EM structures of HCAR2 bound to ligands GSK256073, Acipimox [2],
MK-1903, and SCH900271 [23] have been reported. These works provided valuable
insights into ligand recognition and receptor activation mechanism of HCAR3 and
HCAR2. However, the structural features and ligand preferences of HCAR3 and
HCAR2 required more data support, which could accelerate high-affinity HCAR3
selective drug development.

Here, we resolved the cryo-EM structures of human HCAR3-Gi bound with selective agonists 6O, PLA, and IBC293, as well as the structures of human HCAR3-Gi
and HCAR2-Gi bound with the non-selective agonist acifran. In combination with
structural analysis, molecular dynamics (MD), and mutagenesis results, these
structures revealed the ligand recognition and selectivity mechanisms of HCAR3 and
HCAR2. We also analyzed the active mechanism of HCAR3 compared to inactive
HCAR2. The structural information provided a rational foundation for HCAR3-specific
agonist optimization and design.

## Results

### Overall structures of HCAR3-Gi complexes and HCAR2-Gi complexes

To acquire stable HCAR3 and HCAR2 complexes, we co-expressed receptors with
Gαi1, Gβ1, and Gγ2 heterotrimer in Sf9 insect cell lines. These complexes were further purified with agonists 6O, PLA, IBC293, and acifran, and assembled with stabilizing antibody scFv16. Through cryo-EM data collection and analysis, five electron
density maps of HCAR3 and HCAR2-Gi1-scFv16 complexes were obtained, with resolutions of 3.31 Å (6O-HCAR3), 3.05 Å (PLA-HCAR3), 3.26 Å (IBC293-HCAR3), 3.18
Å (acifran-HCAR3), and 2.72 Å (acifran-HCAR2) (Figs 1A and S1–S5). The electron
density of ligands, orthosteric pocket, Gi protein interface, and other domains displayed high quality. Accurate atomic models were constructed based on clear density
maps (Fig 1B–1F). Map to model FSC curves indicated that models fit maps well,
providing detailed atomic information for interaction analysis (S6 Fig). HCAR3 and
HCAR2 structures exhibited canonical seven transmembrane helical topology, similar
to other class A GPCRs [24–27], while extracellular N-terminus (0–16), helix VIII
(301–310), and intracellular C-terminus (311–387) were absent from electron density
maps, indicating their high flexibility.

### HCAR3 ligand binding pocket

Together with ECL1 and ECL3, ECL2 and N-terminus stacked tightly to almost
completely cover the orthosteric binding pocket in HCAR3, isolating the pocket from
extracellular solvent (S7A Fig). This feature was also observed in previous HCAR2
structures [2,28]. Four agonists fit well into orthosteric binding pocket, while agonists
exhibited distinct chemical structures except for a conserved carboxyl moiety (Figs
2A and S7B). Conserved interactions between four agonists and HCAR3 included
the polar interaction of conserved carboxyl moiety with Y284[7.43], as well as the
π–π stacking of agonists' aromatic moieties with F107[3.32], suggesting F107[3.32] and
Y284[7.43]'s essential roles in agonist recognition and receptor activity regulation. Salt

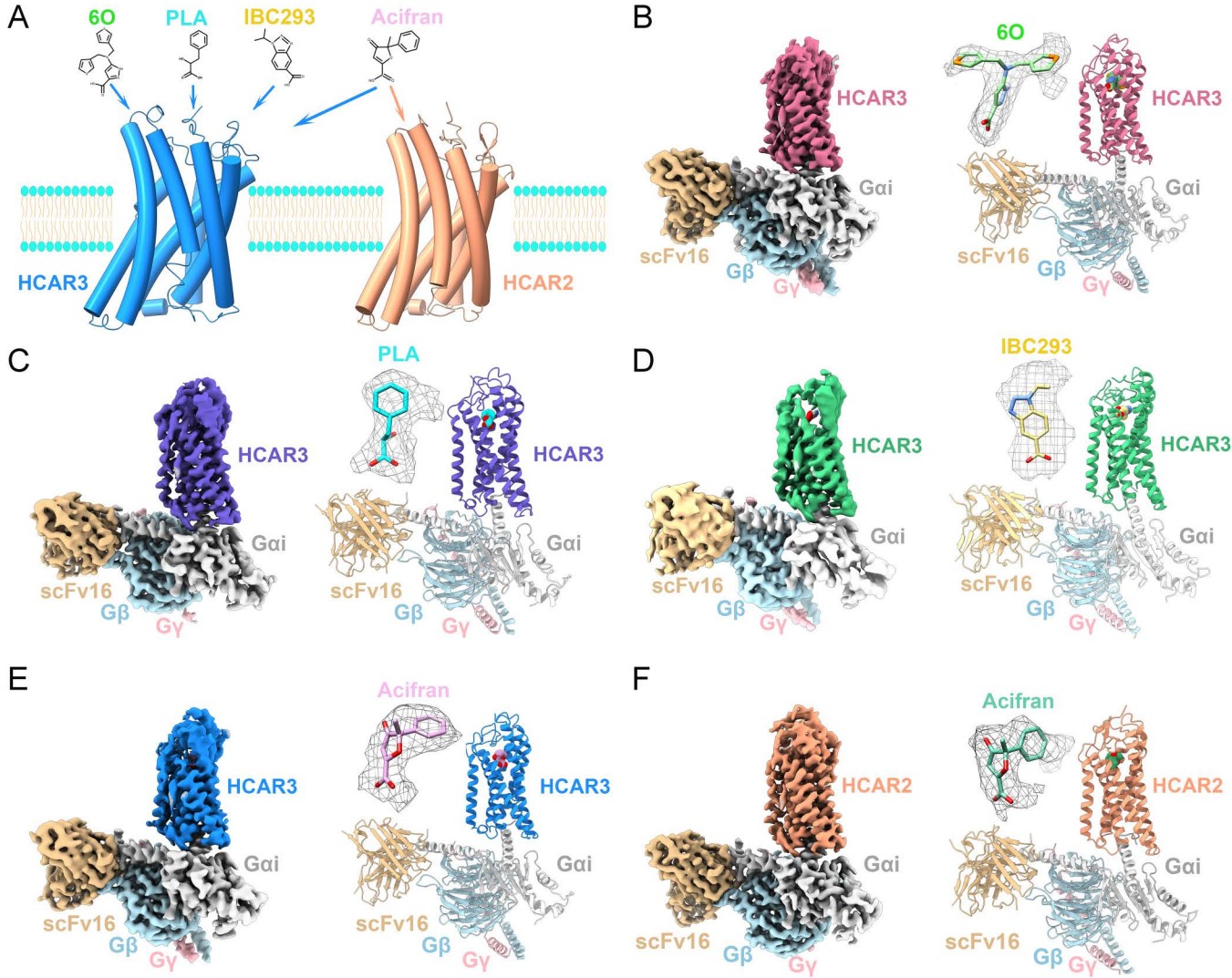

**Fig 1. Cryo-EM structures of agonist-bound HCAR3-Gi1 and HCAR2-Gi1 complexes. (A)** Schematic illustration of agonist selectivity to HCARs.
**(B-F)** Cryo-EM maps and models in cartoon performance of HCAR3-Gi1 complexes in the presence of 6O **(B)**, PLA **(C)**, IBC293 **(D)**, acifran **(E)**, and
HCAR2-Gi1 complex in the presence of acifran **(F)**. Agonists are shown in stick representation, and corresponding electron densities are depicted
as gray meshes. The maps and structural models are colored by subunits: Pale violet red, 6O-HCAR3; slate blue, PLA-HCAR3; medium sea green,
IBC293-HCAR3; dodger blue, acifran-HCAR3; light salmon, acifran-HCAR2; light gray, Gαi; light blue, Gβ; light pink, Gγ; burly wood, scFv16; light green,
6O; cyan, PLA; khaki, IBC293; plum, acifran (HCAR3); medium aquamarine, acifran (HCAR2).

bridges were observed between carboxyl groups of 6O, PLA, acifran, and R111$^{3.36}$. Moreover, S179$^{45.52}$ formed hydrogen
bonds with 6O, PLA, and acifran, while IBC293 lacked interactions with R111$^{3.36}$ and S179$^{45.52}$, indicating that R111$^{3.36}$ and
S179$^{45.52}$ were probably neither conserved nor essential for ligand recognition in HCAR3. This phenomenon differed from
previous HCAR2 articles which highlighted conserved R111$^{3.36}$ in ligand recognition and receptor activation [28,29]. Exten-
sive hydrophobic contacts and van der Waals interactions were observed between hydrophobic residues (V83$^{2.60}$, W93$^{23.50}$,
V103$^{3.28}$, L104$^{3.29}$) and thiophene group of 6O, benzene group of PLA and acifran, isopropyl group, and aromatic dual-ring
of IBC293, respectively, facilitating better fit of ligands into orthosteric pocket (Figs 2B and S8A–S8D). The additional thio-
phene group interacted with L30$^{1.35}$, L34$^{1.39}$, F277$^{7.36}$, and L280$^{7.39}$, enabling 6O to bind more stably in the pocket

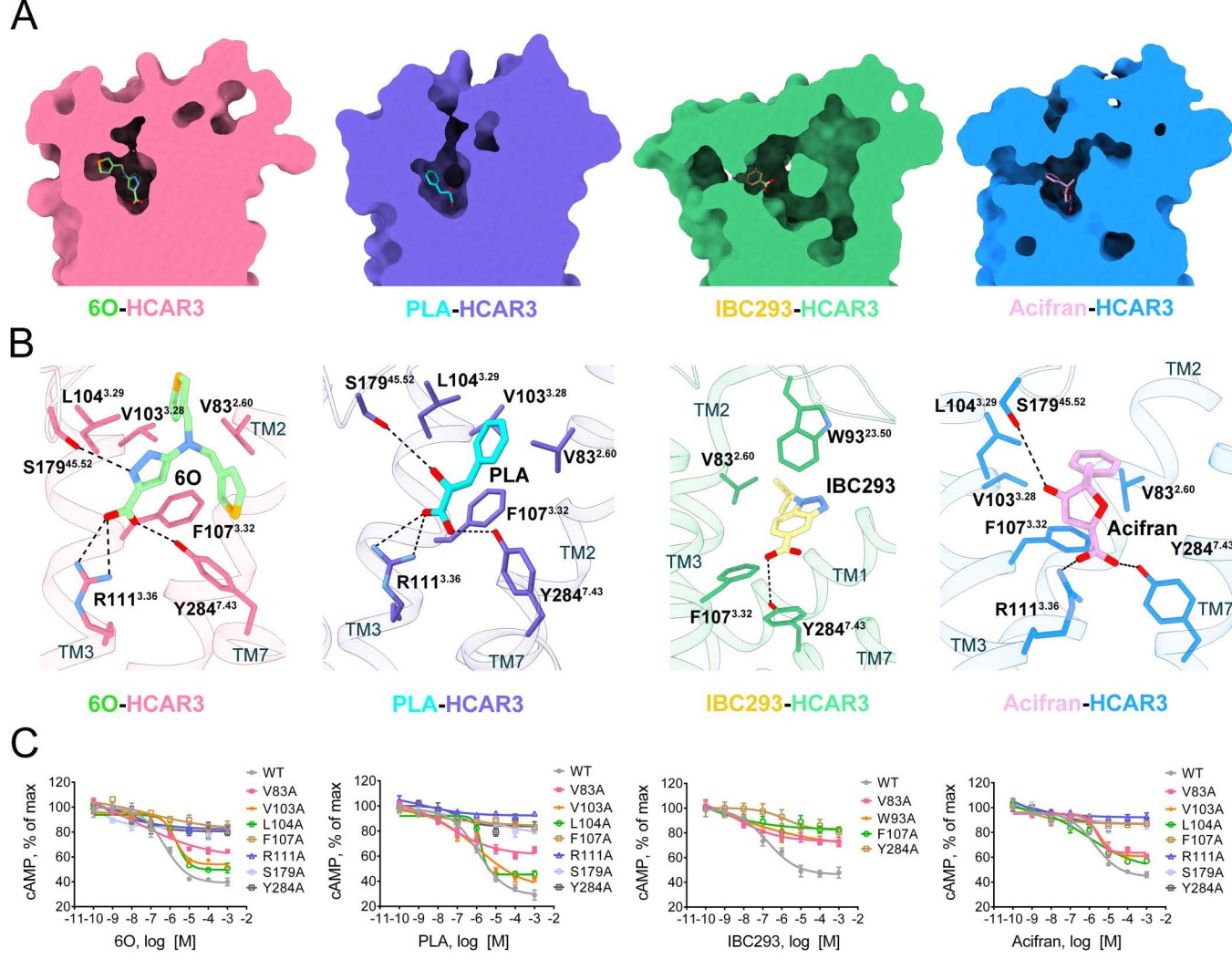

**Fig 2. Orthosteric binding pocket of active HCAR3 in the 6O-, PLA-, IBC293-, and acifran-bound forms. (A)** Vertical cross-section of HCAR3 binding pockets in the presence of 6O, PLA, IBC293, and acifran. **(B)** Key residues that interact with 6O (light green), PLA (cyan), IBC293 (khaki), and acifran (plum) in HCAR3. **(C)** Concentration-response curves of cAMP assays for HCAR3 mutants with a single point mutation of key residues. The data are presented as means±SEM. The experiments are performed in triplicates. The underlying data for this figure can be found in S1 Data.

(S9A Fig). The essential roles in ligand recognition and receptor activation of mentioned residues were validated through single mutation, among which F107[3.32]A and Y284[7.43]A mutants significantly decreased HCAR3 activation, while cell expression levels of mutants were comparable to that of wild type (Figs 2C and S9B–S9D). Furthermore, different ligand-interacting amino acid compositions suggested that HCAR3 adopted different recognition patterns for the four ligands.

## Recognition Mechanism of HCAR3 for Different Agonists

To better understand the affinity and binding pose differences of 6O, PLA, IBC293, and acifran, we aligned four ligand-bound HCAR3 structures. All four agonists were located in HCAR3 orthosteric binding pocket (S10A Fig). Notably, two

hydrophobic regions were observed in HCAR3 orthosteric binding pocket, including region 1 (R1) constituted by TM2 and TM3 residues, as well as region 2 (R2) constituted by TM1 and TM7 residues (Fig 3A).

Surface plasmon resonance assays elucidated that highly subtype-specific agonist 6O possessed the highest efficacy among these four ligands (S11A–S11D Fig). To explain why compound 6O bound HCAR3 with the highest affinity, we compared 6O-bound and acifran-bound HCAR3 structures. The benzene moiety of acifran simultaneously formed π–π interactions with F107$^{3.32}$ and inserted shallowly into R1. Unlike acifran, the central pyrazole moiety of 6O interacted with F107$^{3.32}$ through π–π stacking, with two thiophene groups occupying both hydrophobic regions R1 and R2 (Figs 3B, 3C, and S10B). When focusing on R1, the thiophene moiety of 6O fitted deeply into R1, causing a 1.1–2.5 Å movement of

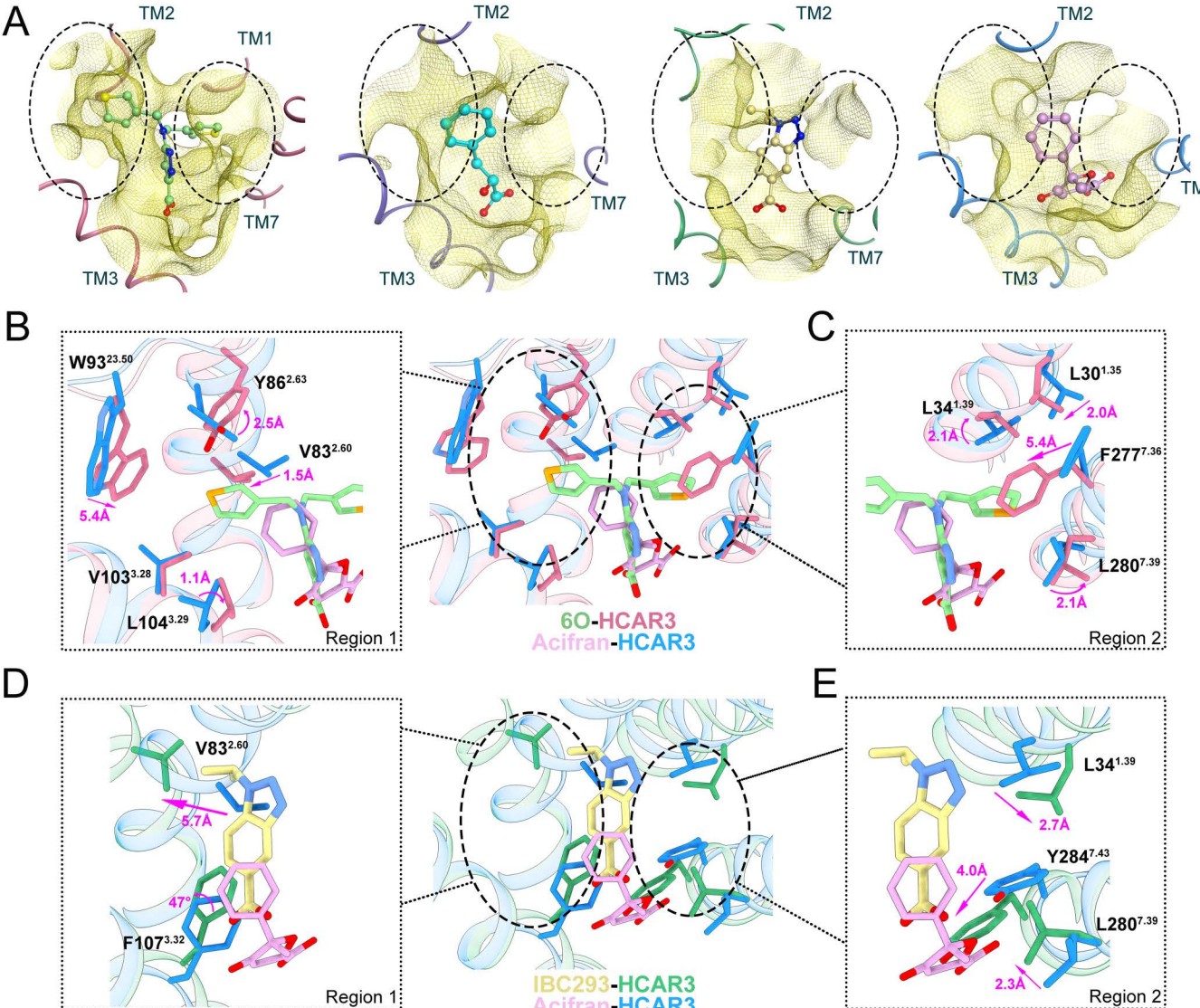

**Fig 3. Comparison of R1 and R2 conformations in 6O-, PLA-, IBC293-, and acifran-HCAR3 structures. (A)** Surface representation of R1 (large circle) and R2 (small circle) conformations. The structures and agonists are colored differently. Light green and pale violet red, 6O-HCAR3; cyan and slate blue, PLA-HCAR3; khaki and medium sea green, IBC293-HCAR3; plum and dodger blue, acifran-HCAR3. **(B, C)** Detailed comparison of R1 and R2 residues between 6O- and acifran-HCAR3 structures. **(D, E)** Detailed comparison of R1 and R2 residues between IBC293- and acifran-HCAR3 structures.

V83$^{2.60}$, Y86$^{2.63}$, and V103$^{3.28}$, as well as a remarkable 5.4 Å shift of W93$^{23.50}$ through hydrophobic interactions (Fig 3B). Movement of these residues resulted in the closure of R1 in 6O-HCAR3 structure (S10C Fig). Moreover, another thiophene moiety of 6O formed hydrophobic interactions with L30$^{1.35}$, L34$^{1.39}$, and L280$^{7.39}$ in R2, causing a 2.0–3.0 Å displacement. Together with 5.4 Å downward movement of F277$^{7.36}$, R2 formed a groove-like structure and accommodated thiophene group tightly, similar to R1 (Figs 3C and S10D). In summary, 6O's high affinity towards HCAR3 arose from extensive hydrophobic interactions and full occupation of R1 and R2 of the orthosteric binding pocket, acifran only occupied R1.

Unlike binding poses of other three agonists, IBC293 adopted a holistic planar pose in the binding pocket. Compared to acifran, the backbone of IBC293 consisted of an aromatic dual ring (benzotriazole), with considerable rigidity. Stronger rigidity prevented carboxyl group of IBC293 from generating salt bridge with R111$^{3.36}$, while acifran could bend to facilitate the polar interaction between its carboxyl group and R111$^{3.36}$ (S10E Fig). Consistently, cAMP assay results indicated that R111$^{3.36}$A mutation barely interfered HCAR3 activation by IBC293, while the mutation nearly abolished the binding of acifran (Figs 2C and S10F). In contrast to acifran, benzotriazole of IBC293 formed π–π interactions with F107 $^{3.32}$ in R1, eliciting the rotation of F107$^{3.32}$ (Figs 3D and S10B). V83$^{2.60}$ shifted outward by 5.7 Å due to the isopropyl moiety of IBC293, producing a loosely packed R1 (Figs 3D and S10G). Furthermore, Y284$^{7.43}$ shifted 4.0 Å to better interact with carboxyl group of IBC293 in R2. The 2.3–2.7 Å displacement was also observed in residues L34$^{1.39}$ and L280$^{7.39}$, through hydrophobic interaction (Fig 3E). To sum up, IBC293 possessed a distinct binding pose attributable to its distinct backbone and interaction with F107$^{3.32}$.

## Mechanism of ligand selectivity between HCAR3 and HCAR2

In the previous section, we analyzed the differences in the mechanisms of HCAR3 ligand recognition and their effects on binding affinity. Notably, IBC293, 6O and PLA selectively bound to HCAR3 but not to HCAR2, prompting further investigation into the molecular basis of this specificity.

We first aligned reported HCAR3 structures bound to ligand containing cyclic chemical moieties (niacin, compound 5c, acifran, IBC293, 6O, PLA), to summarize features of HCAR3 ligand recognition (Fig 4A) [21]. The alignment revealed critical residues involved in ligand binding, including F107$^{3.32}$, R111$^{3.36}$, S179$^{45.52}$, Y284$^{7.43}$. Niacin, PLA, 6O, and acifran adopted similar binding pose, with their carboxyl groups interacting with R111$^{3.36}$ and Y284$^{7.43}$. Interestingly, IBC293 adopted a "wedge-like" lateral orientation, inserting into R1, and its carboxyl group lacked polar interaction with R111$^{3.36}$. However, R111$^{3.36}$ interacted with all ligands in reported HCAR2 structures. Therefore, we compared acifran-HCAR2 and acifran-HCAR3 structures (~100-fold affinity difference, S11D and S12A Figs), and observed similar ligand-receptor interactions and hydrophobic grooves (R1), while the binding pocket of HCAR3 was larger (Figs 4B, S12B, and S13). In HCAR2, L83$^{2.60}$ (V83$^{2.60}$ in HCAR3) and M103$^{3.28}$ (V103$^{3.28}$) had large side chains and constricted the volume of R1 (S12C Fig). W93$^{23.50}$, N86$^{2.63}$ (Y86$^{2.63}$) and W91$^{23.48}$ (S91$^{23.48}$) were oriented toward acifran, causing R1 to fit tightly around the ligand (S12D Fig). Moreover, L107$^{3.32}$ (F107$^{3.32}$) played a crucial role in ligand recognition of HCAR2 and HCAR3. The benzene ring of acifran underwent a 64.5° rotation induced by its π–π conjugation with F107$^{3.32}$ in HCAR3, which facilitated deeper insertion of acifran's benzene moiety into the R1 region in HCAR2 than in HCAR3 (S12C Fig). Homology mutation experiments validated the importance of these residues (S12E Fig). MD simulations also indicated that acifran possessed higher stability in HCAR2 pocket than in HCAR3's (S12F Fig).

The HCAR2 pocket was notably smaller than that of HCAR3, which likely explained previously reported 100- to 1,000-fold affinity difference of IBC293, 6O, and PLA to HCAR3 compared to HCAR2. In HCAR2, N86$^{2.63}$ formed polar interactions with W91$^{23.48}$, causing W91$^{23.48}$ to act like a lid over R1, thereby reducing R1 size and stabilizing TM2 conformation. Additional nitrogen-carbon linker of 6O required more flexible TM2 extracellular side and loosely packed R1 to accommodate, which likely resulted in clashes with L83$^{2.60}$ and N86$^{2.63}$ of HCAR2 (Fig 4C). Similar clashes were also observed in alignment between IBC293-HCAR3 and acifran-HCAR2, because of IBC293's rigidity (Fig 4D). Additionally, R2, which was

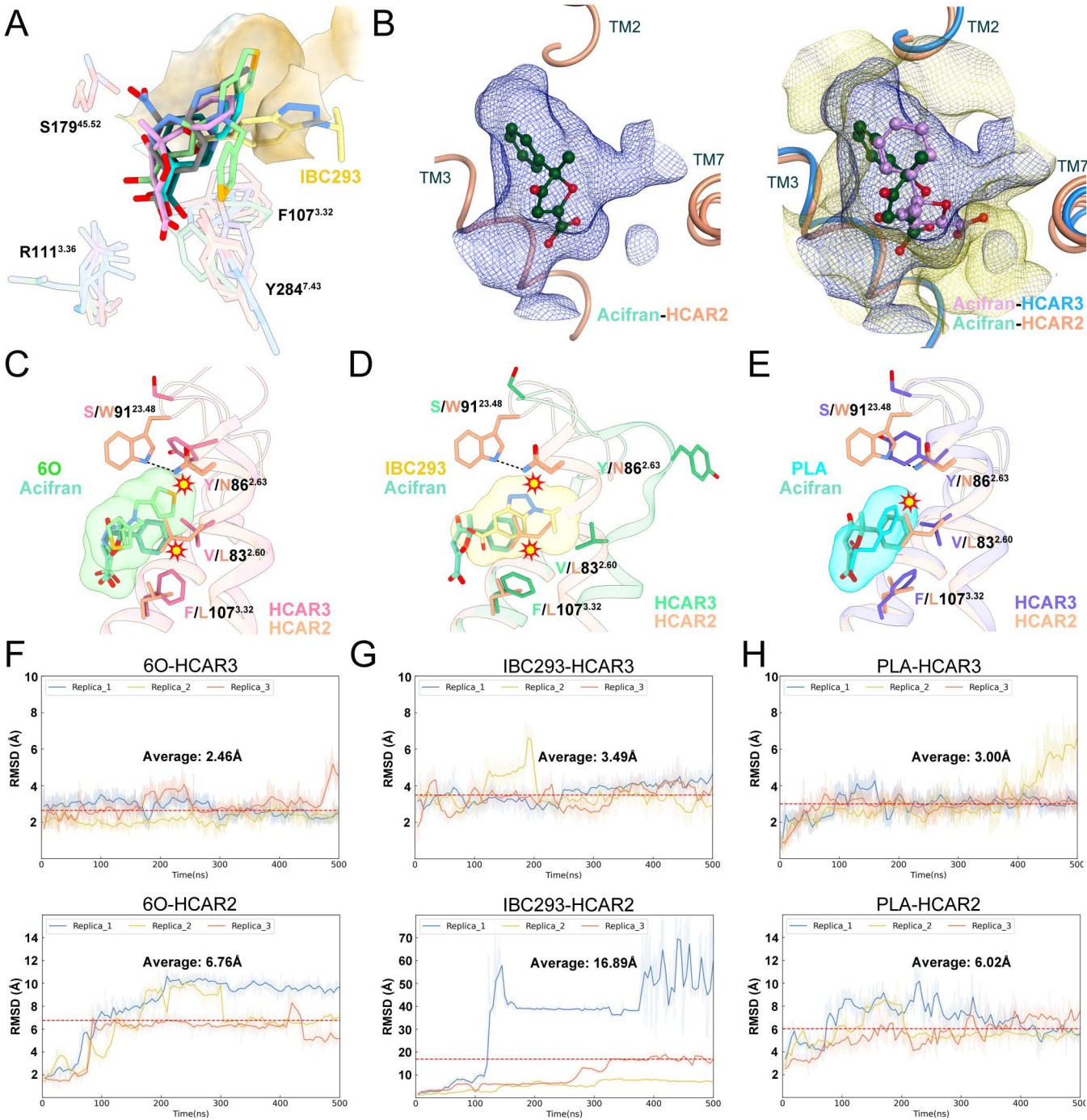

**Fig 4. Structural basis of ligand selectivity between HCAR3 and HCAR2. (A)** Superimposed HCAR3 agonists niacin (PDB: 9JID), compound 5c (PDB: 8JEI), acifran, IBC293, 6O, PLA, and ligand-interacting residues. **(B)** Surface representation of HCAR2 orthosteric pocket, as well as superimposed HCAR2 and HCAR3 orthosteric pockets. The structures and agonists are colored differently. Medium aquamarine and light salmon, acifran-HCAR2; plum and dodger blue, acifran-HCAR3. Blue, HCAR2 surface; yellow, HCAR3 surface. **(C–E)** Potential clash of HCAR2 residues with 6O, IBC293, PLA. **(F–H)** Ligand stability evaluation in HCAR2 and HCAR3. RMSD value Raw (light color) and smoothed (10 ns moving average, dark color). Red dashed line is the average RMSD value for three replica trajectories.

occupied by a thiophene group of 6O, was absent in HCAR2 orthosteric pocket. These two phenomena may be attributable to HCAR2's inability to accommodate large-size 6O (S14A Fig). A potential clash with L83$^{2.60}$ was also observed in PLA-HCAR3 (Fig 4E). Furthermore, 6O, IBC293, and PLA formed π–π interaction with F107$^{3.32}$ in HCAR3, whereas the corresponding L107$^{3.32}$ in HCAR2 lacked, reducing ligand stability. Homology mutation experiments further supported the importance of residue differences on 83$^{2.60}$, 86$^{2.63}$, 91$^{23.48}$, 107$^{3.32}$ (S14B–S14D Fig). MD simulations revealed differences in the RMSD of 6O (Fig 4F), IBC293 (Fig 4G), and PLA (Fig 4H) when bound to HCAR3 versus HCAR2, indicating that these three ligands exhibited higher stability in the orthosteric pocket of HCAR3 compared to HCAR2.

In summary, the interaction of residue 107$^{3.32}$ (F/L) with ligands influenced their binding affinity, while size differences of orthosteric pockets, formed by residues 83$^{2.60}$ (V/L), 86$^{2.63}$ (Y/N), and 91$^{23.48}$ (S/W) contributed to 6O, IBC293, PLA's selectivity to HCAR3, not HCAR2.

## Activation mechanism of HCAR3

Until now, no inactive HCAR3 structure has been reported. Since HCAR3 and HCAR2 share 96% sequence homology, inactive state HCAR2 structure (PDB ID: 7ZL9) was used to analyze HCAR3 activation mechanism [29].

We aligned the structures of 6O-HCAR3, PLA-HCAR3, and IBC293-HCAR3 with the inactive-state HCAR2 structure for comparative analysis (S15A Fig). Relative to inactive HCAR2, these agonist-bound HCAR3 structures exhibited distinct outward displacements in their transmembrane helices: TM7 outward shifted ~6.8 Å (6O-HCAR3), 5.7 Å (PLA-HCAR3), and 3.3 Å (IBC293-HCAR3) (S15B Fig). Notably, the TM7 displacement created a structural niche that accommodated the Gi α5 helix insertion, thereby enabling HCAR3-Gi complex formation, while the absence of this niche in inactive HCAR2 would result in potential steric clashes between TM7 and Gi α5 helix (S15C Fig).

Further analysis revealed additional helical rearrangements: In TM5, the extracellular ends displaced ~8.5 Å (6O-HCAR3), 6.7 Å (PLA-HCAR3), and 3.4 Å (IBC293-HCAR3), while intracellular ends shifted ~5.3 Å, 5.1 Å, and 6.0 Å, respectively (S16A Fig). Similarly, TM6 exhibited extracellular displacements of ~2.6 Å, 0.7 Å, and 1.8 Å, paired with intracellular shifts of ~1.9, 1.8, and 0.9 Å (S16B Fig). These concerted movements suggested a global conformational reorganization upon agonist binding.

To investigate the allosteric transmission of activation signal from orthosteric binding pocket towards G protein coupling interface, "CWxP" motif, which is located near ligand binding pocket and contains the "toggle switch" residue, was first analyzed. Interestingly, F244$^{6.48}$ replaced W$^{6.48}$ in HCAR3 and HCAR2, which is common in δ-branch GPCRs (including HCAR family) [30]. Compared to inactive HCAR2, F244$^{6.48}$ adopted moderate displacement together with C243$^{6.47}$ and P246$^{6.50}$ from "CWxP" motif (S16C Fig). Similarly, the "PIF" motif, consisting of P200$^{5.50}$, I115$^{3.40}$, and F240$^{6.44}$, underwent moderate residue movement caused by agonist binding (S16D Fig). Notably, moderate conformational changes in "CWxP" and "PIF" motifs, located at the base of orthosteric pocket, "amplified" activation signal and induced larger conformational rearrangements in "NPxxY" and "DRY" motifs, located at G protein interface. In HCAR3 and HCAR2, D290$^{7.49}$ replaced N$^{7.49}$ in "NPxxY" motif. This replacement occurs in 18% of class A GPCRs based on sequence alignment. The difference between aspartic acid and asparagine is probably related to different downstream protein recruitment [31]. The other two residues in "NPxxY" motif, P291$^{7.50}$ and Y294$^{7.53}$, rotated along helical axis, resulting in shift of TM7 cytoplasmic end to better accommodate G protein coupling (S16E Fig). Furthermore, large rotameric shift of Y294$^{7.53}$ generated electrostatic interaction with R125$^{3.50}$ from "DRY" motif in 6O- and PLA-bound HCAR3, breaking ionic lock between D124$^{3.49}$ and R125$^{3.50}$, which stabilized inactive conformation in inactive HCAR2 [32]. Interestingly, the displacement of Y294$^{7.53}$ and R125$^{3.50}$ was not significant in IBC293-bound HCAR3 (S16F Fig). Holistically, compared to inactive HCAR2, key motifs underwent similar displacements in 6O-, PLA-, and IBC293-bound HCAR3 and HCAR2, with the motif densities being well resolved (S17 Fig), suggesting a typical class A GPCR activation mechanism adopted by HCAR3. These conformational changes opened up the transmembrane domain (TMD) cavity to allow the insertion of α5 helix of Gαi into the TMD core of HCAR3.

PLOS Biology

## Discussion

HCAR3 is expressed in adipocytes and immune cells, making it a potential drug target for lipid metabolism and immune regulation. Here, we determined four cryo-EM structures of HCAR3-Gi complexes with highly selective agonists 6O, PLA, IBC293, and the less selective agonist acifran. The cryo-EM structures reveal the occupation to two hydrophobic regions R1 and R2 of HCAR3's orthosteric pocket constitutes the primary determinant for 6O's high-affinity binding to HCAR3. We have also elucidated the mechanism underlying the distinctive binding pose of IBC293 to HCAR3, unlike other HCAR3 ligands. Besides, the mechanism of the inability of HCAR2 to recognize HCAR3-specific agonists compound 6O, PLA, and IBC293. F/L107$^{3.32}$ are the most critical residues determine the ligand selectivity between HCAR2 and HCAR3, and V/L83$^{2.60}$, Y/N86$^{2.63}$, S/W91$^{23.48}$ determine the pocket size divergence between these two receptors (S18 Fig).

Interestingly, residues at 3.32 position were mostly reported to play critical roles in ligand recognition in other class A GPCRs. Most aminergic receptors have conserved D$^{3.32}$ forming polar interactions with ligands to stabilize ligand binding [33]. Nonpolar residues$^{3.32}$, for example, F$^{3.32}$ and M$^{3.32}$ in prostaglandin receptors DP2 and EP2, respectively, participate in hydrophobic binding pocket formation and hydrophobic interactions with ligands [34,35]. However, it is rarely reported that residue$^{3.32}$ difference in homologous receptor families could affect ligand selectivity. Our results revealed that the F/L107$^{3.32}$ in HCAR3/HCAR2 had dual functions, both in ligand recognition and ligand selectivity, providing supplementary insights into how specific sites influence ligand binding in class A GPCRs.

In this study, we reported the affinity differences of acifran to HCAR3-Gi and HCAR2-Gi complexes. The underlying mechanism depended on the residue differences in the R1 regions of orthosteric binding pockets, especially residue 107 (F/L) that affected ligand-receptor interactions and residues, 83 (L/V), 86 (N/Y), 91 (W/S) that affected pocket size. Suzuki and colleagues recently reported HCAR3 and HCAR2 structures with acifran binding as well [22]. We emphasized map comparison between their acifran-HCAR3-Gi complex (PDB ID: 8IHJ) and ours. The electron density of residues related to selectivity and ligand acifran is clearer in our map, facilitating determination of the benzene ring orientation (S19A Fig). Moreover, our map had a more complete density for ECL1 and ECL2 hairpin regions, allowing us to better define side chains (S19B and S19C Fig). In addition to residues 86$^{2.63}$, 91$^{23.48}$, 103$^{3.28}$, and 178$^{45.51}$, we especially elucidated the structural mechanism for the close relationship between acifran selectivity and residue differences at sites 83$^{2.60}$, 86$^{2.63}$, 91$^{23.48}$, 107$^{3.32}$. Our findings provide valuable and progressive supplements to results reported by Suzuki and colleagues.

Overall, our study provides insights into designing HCAR3 high-affinity, high-selectivity ligands, which could help develop low side-effect drugs that target HCAR3 without targeting HCAR2.

## Materials and methods

### Cloning and purification of the HCAR3-Gi1 and HCAR2-Gi1 complex

The pFastbac vector (Gibco) was harnessed to construct plasmids for the expression of the entire length of human HCAR3 and HCAR2 proteins, which were modified at the N-terminus with a hemagglutinin signal sequence, a FLAG epitope, a 3C protease cleavage site, and were tagged at the C-terminus with a His tag. Additionally, a variant of Gαi1 exhibiting dominant-negative behavior (DNGαi1) [36], distinguished by G203A and A326S mutations, was engineered, retaining the same modifications, except for the FLAG tag, as HCAR3 and HCAR2. Utilizing the Bac-to-Bac baculovirus expression system, these constructs, in conjunction with Gβ1γ2, were simultaneous expression in Spodoptera frugiperda Sf9 cells (Invitrogen). In the co-expression protocols, Sf9 cells, cultured in suspension at a concentration of $4 \times 10^6$ cells/ml, received co-infection from baculoviruses carrying the genes for HCAR2 or HCAR3, DNGαi1, and Gβ1γ2 in a 10:10:1 mixture. The Sf9 cells were collected 48 hours after infection by centrifugation at 4,000$g$ for 10 min, and the cell pellets were then preserved at −80 °C.

For the extraction of the HCAR3-DNGαi1-Gβ1γ2 and HCAR2-DNGαi1-Gβ1γ2 complexes with various ligands, specific agonists were introduced at different stages of the purification process. The HCAR3-DNGαi1-Gβ1γ2 complex was

targeted with 100 µM of acifran (APEXBIO B6848), complemented by 6O at 16 µM (Wuxi AppTec), IBC293 at 75 µM (Tocris 2469), and PLA at 100 µM (Aladdin 7326-19-4), acting as agonists. Similarly, the HCAR2-DNGαi1-Gβ1γ2 complex was subjected to the same acifran concentration. A buffer containing 0.5 mM EDTA and 10 mM Tris (pH 7.5) was used to resuspend and lyse the cell pellets, with magnetic stirring at 4°C for an hour to facilitate the complexes' assembly. Following this, cell membranes were isolated by centrifugation and resuspended in a solubilization buffer consisting of 20 mM HEPES (pH 7.5), 100 mM NaCl, 10% (w/v) glycerol, 10 mM MgCl$_2$, 5 mM CaCl$_2$, 1 mM MnCl$_2$, 100 µg/ml benzamidine, 0.2 µg/ml leupeptin, 25 µU/ml apyrase (NEB), and 100 µU/ml lambda phosphatase (NEB). A solution of 1% (w/v) n-dodecyl-β-D-maltoside (DDM) and 0.1% (w/v) cholesteryl hemisuccinate (CHS) was then added, and the mixture was incubated at 4 °C for 2 hours.

The supernatant obtained post-centrifugation was subjected to a 1-hour incubation at 4 °C with Anti-Flag M1 antibody affinity resin (Sigma Cat# A4596). For washing, a NH buffer composed of 20 mM HEPES (pH 7.5) and 100 mM NaCl, enhanced with 0.1% DDM, 0.01% CHS, and 2 mM CaCl$_2$, was employed. Following the wash, a gradual procedure was applied to transition the M1 resin to a medium containing 0.1% (w/v) LMNG. Subsequently, a 10 × CMC buffer, which includes the NH buffer with added 0.01% LMNG, 0.001% CHS, and 2 mM CaCl$_2$, was utilized for a gentle final wash of the M1 resin.

Following the washing and buffer exchange processes, the HCAR3-Gi1 and HCAR2-Gi1 complexes were retrieved from the M1 resin through elution with a buffer comprising NH buffer, 5 mM EDTA, 0.00075% LMNG, 0.00025% (w/v) glycol-diosgenin (GDN), 0.0001% CHS, and 200 µg/ml of Flag peptide.

The eluted protein was concentrated using an ultrafiltration tube (Merck Amicon Ultra-15ML), followed by the addition of the antibody fragment scFv16 to the complex. This mixture was then incubated for a minimum of 2 hours on ice, maintaining a 1:1.5 ratio [37]. To achieve further purification of the HCAR2-Gi1-scFv16 and HCAR3-Gi1-scFv16 complexes, a Superdex 200 Increase 10/300 column (GE Healthcare), previously equilibrated with 10 × CMC buffer, was employed.

### Cryo-grid preparation and EM data collection

Initially, a 100 Holey Carbon film grid (Au, 300 mesh, N1-C14nAu30-01) was treated with a Tergeo plasma cleaner to ensure its pre-discharge. Following this preparation, the HCAR3-Gi1-scFv16 and HCAR2-Gi1-scFv16 complexes were applied to the grid. The application process included blotting the grids for 3.5 seconds in an environment controlled at 10 °C and 100% humidity, immediately followed by rapid freezing in liquid ethane using the Vitrobot I Freezing plunger (Thermo Fisher Scientific).

The collection was carried out using a Titan Krios Gi3 microscope at 300 kV (Thermo Fisher Scientific FEI). Movies were recorded at a magnification of 105,000× using a Gatan K3 BioQuantum imaging system, with the pixel size set to either 0.85 Å or 0.83 Å. The inelastically scattered electrons were filtered out using a GIF-quantum energy filter (Gatan, USA). Data collection involved capturing movie stacks with a defocus range between −1.1 and −2.0 µm, utilizing a 20 eV slit width. Each sample had 50 frames collected over a total exposure time of 2.0 seconds, with the electron dose rate of 21.2 electrons per pixel per second. The acquisition of single-particle data was facilitated by the SerialEM 3.7 software.

### Image processing and 3D reconstructions

Cryo-EM datasets for the HCAR3-Gi1-scFv16 and HCAR2-Gi1-scFv16 complexes, bound to various agonists, were analyzed using cryoSPARC v4.1.1 [38]. For the HCAR3 complex bound to 6O, a total of 3,159 movie stacks underwent alignment for motion correction and CTF (Contrast Transfer Function) estimation. Template picking resulted in the extraction of 2,572,904 particles, which were then narrowed down to 509,850 particles following 2D classification. Subsequent 3D classification and ab initio 3D model generation further refined the particle count to 318,073. The processing sequence, which included homogeneous refinement, non-uniform refinement, and local refinement stages, culminated in the production of a map. This map achieved a global resolution of 3.31 Å, determined at an FSC of 0.143.

For the HCAR3 complex bound to PLA, a total of 3,780 movie stacks underwent alignment for motion correction and CTF (Contrast Transfer Function) estimation. Template picking resulted in the extraction of 3,100,214 particles, which were then narrowed down to 355,650 particles following 2D classification. Subsequent 3D classification and ab initio 3D model generation further refined the particle count to 257,140. The processing sequence, which included homogeneous refinement, non-uniform refinement, and local refinement stages, culminated in the production of a map. This map achieved a global resolution of 3.05 Å, determined at an FSC of 0.143. For the HCAR3 complex bound to IBC293, a total of 3,249 movie stacks underwent alignment for motion correction and CTF (Contrast Transfer Function) estimation. Template picking resulted in the extraction of 2,498,842 particles, which were then narrowed down to 505,659 particles following 2D classification. Subsequent 3D classification and ab initio 3D model generation further refined the particle count to 268,826. The processing sequence, which included homogeneous refinement, non-uniform refinement, and local refinement stages, culminated in the production of a map. This map achieved a global resolution of 3.26 Å, determined at an FSC of 0.143. For the HCAR3 complex bound to acifran, a total of 3,258 movie stacks underwent alignment for motion correction and CTF (Contrast Transfer Function) estimation. Template picking resulted in the extraction of 2,515,469 particles, which were then narrowed down to 411,277 particles following 2D classification. Subsequent 3D classification and ab initio 3D model generation further refined the particle count to 375,522. The processing sequence, which included homogeneous refinement, non-uniform refinement, and local refinement stages, culminated in the production of a map. This map achieved a global resolution of 3.18 Å, determined at an FSC of 0.143. For the HCAR2 complex bound to acifran, a total of 3,542 movie stacks underwent alignment for motion correction and CTF (Contrast Transfer Function) estimation. Template picking resulted in the extraction of 3,059,535 particles, which were then narrowed down to 841,066 particles following 2D classification. Subsequent 3D classification and ab initio 3D model generation further refined the particle count to 877,365. The processing sequence, which included homogeneous refinement, non-uniform refinement, and local refinement stages, culminated in the production of a map. This map achieved a global resolution of 2.72 Å, determined at an FSC of 0.143.

## Model building and refinement

For the construction of the HCAR3-Gi1-scFv16 and HCAR2-Gi1-scFv16 complex models, the HCAR3 and HCAR2 templates were sourced from Alphafold (https://alphafold.ebi.ac.uk/), while the Gi-scFv16 portion was modeled on the structure available from the FPR2-Gi Cryo-EM structure (PDB ID: 6OMM) [39]. The preliminary model docking into the electron density maps was executed using UCSF Chimera, followed by detailed manual adjustments and structural refinements with COOT [40]. Further real-space refinement was conducted using Phenix [41]. The Phenix suite also facilitated the statistical analysis of the refined models. Visual representations of the molecular structures were generated utilizing software tools such as UCSF Chimera [42], UCSF ChimeraX [43,44], and PyMOL (https://www.schrodinger.com/pymol).

## cAMP assay

Full-length variants of HCAR3 and HCAR2, including specific point mutations, were incorporated into pcDNA3.1 vectors for conducting cAMP assays. The cAMP levels were quantified using the cAMP-Gi kit from Perkin Elmer (model TRF0263), adhering to the manufacturer's guidelines. HEK-293 cells (ATCC CRL-1573) were plated in 24-well plates, achieving a confluency of 70–90% per well. Cells were transiently transfected with either wild-type or mutant HCAR3 plasmids (at a dosage of 500 ng per well in a 24-well format) using Lipofectamine 3000 (Invitrogen). After 36 hours, the cells, once rinsed with PBS, were treated with a stimulation buffer,including 500 µM IBMX, and incubated for 30 min in a 384-well plate maintained at 37 °C, with a seeding density of 4,000 cells per well. This step was followed by the addition of 12 µM forskolin to each well, with the incubation continuing for an additional 45 min at the same temperature. Subsequently, each well received 5 µL of the cAMP Eu-cryptate solution and the anti-cAMP-d2 solution, and the mixture was incubated for 1 hour at room temperature [45]. Fluorescence readings at 620/665 nm were then

gathered using a multimode plate reader (Perkin Elmer EnVision 2105) [46]. The acquired data were analyzed using GraphPad Prism version 9.0, with statistical significance set at $p < 0.05$. All experimental procedures were replicated three times to ensure accuracy.

### Flow cytometry analysis

Wild-type HCAR3, HCAR3 mutants, wild-type HCAR2, and HCAR2 mutants in transfected HEK-293 cells were first incubated with 5% (w/v) BSA at room temperature for 15 min for blocking. This step was followed by incubation with an anti-Flag antibody (Thermo Fisher Scientific) at 4°C for one hour. The cells were then rinsed three times with PBS and subsequently exposed to an Alexa-488-conjugated secondary antibody (Beyotime) for an hour at 4 °C, avoiding light exposure. The fluorescence intensity was measured using a BD Accuri C6 Plus flow cytometer, allowing for the comparison of surface expression levels of HCAR3 mutants against the wild-type HCAR3, which was set as the 100% benchmark. The statistical evaluation of the data was conducted using GraphPad Prism 9.0, applying one-way ANOVA for analysis, with significance thresholds set at $p < 0.05$.

### Surface plasmon resonance measurement

The affinity of agonists for both wild-type HCAR3 and HCAR2 was determined using the Biacore X100 system, which operated with a running buffer composed of 2 mM HEPES (pH 7.4), 10 mM NaCl, and 5% (v/v) DMSO. The amine-coupling method was applied to attach purified wild-type HCAR2 and HCAR3 to the surface of a CM5 sensor chip at a pH of 5.0. For the interaction analysis, agonists at various concentrations were prepared in the running buffer and flowed over the chip for 100 seconds to assess the association phase. Following this, the running buffer was used to flush the chip, facilitating a 50-second dissociation phase for the agonist. The Biacore X100 system continuously recorded sensorgrams, which were subsequently analyzed to deduce the binding affinities ($K_D$) of the interactions.

### Molecule docking

Molecular docking was conducted using Schrödinger Suite (version 2018.4) to construct the PLA-, 6O-, and IBC-HCAR2 complexes. The crystal structure of HCAR2 was obtained from the Protein Data Bank (PDB ID: 8IHI) [22]. The protein structure was prepared by removing water molecules, assigning bond orders, and adding hydrogen atoms. Ligand structures were prepared by generating 3D conformations and ionization states using the LigPrep module. Grid-based docking was conducted with the Glide module, defining the receptor grid based on the initial ligand position in the crystal structure. Glide XP (extra precision) was employed for accurate pose prediction and scoring. The docking result with the best docking score was selected for subsequent MD simulations.

### Molecule dynamics simulations

To examine the stability of different agonists in the binding pocket of HCAR2 and HCAR3. We used MD simulations for PLA-bound, 6O-bound, IBC-bound, acifran-bound HCAR2 and HCAR3 models, where agonists-bound HCAR3 complexes were constructed from this study; acifran-bound HCAR2 complex was constructed from previous reported structures (PDB ID: 8IHI) [22]; PLA-bound, 6O-bound, IBC-bound HCAR2 complexes were constructed from molecular docking. The Membrane Builder module in CHARMM-GUI server [47] was used to prepare the simulation inputs. The agonist-protein complexes were embedded into a membrane bilayer composed of POPC molecules and the position was based on OPM database [48]. The complex was then solvated with TIP3P solvent and 0.15 M NaCl. CHARMM36 forcefield [49] was used for the protein, lipids, and ions. CHARMM general force field (CGenFF) [50] was used for agonists. All MD simulations were performed using GROMACS-2019.4 [51]. The MD simulations were conducted using the default settings specified in the.*mdp* files generated by the CHARMM-GUI server. These default configurations encompassed parameters for

5,000 steps of energy minimization using steepest descent algorithm, two NVT and four NPT equilibration stages, and production run. For the entire simulation, a cutoff distance of 12 Å was applied for both van der Waals interactions and short-range electrostatic interactions. Long-range electrostatic interactions were calculated using the Particle Mesh Ewald algorithm [52]. LINCS algorithm [53] was used to constrain the covalent bonds involving hydrogen atoms. Details about *mdp* file can be found in S2 Table. Three independent MD simulations were conducted for each of the eight agonist-receptor systems. The trajectories were recorded at 100 ps intervals and subsequently analyzed using the MDAnalysis library [54]. Root mean-square deviation (RMSD) calculations were performed using the Cα atoms of the all-helical regions for alignment, and the heavy atoms of the agonists for the RMSD evaluation.

## Supporting information

**S1 Fig. Cryo-EM data processing of HCAR3-Gi1 signaling complex in the compound 6O-bound form. (A)** Size exclusion chromatography profile and SDS-PAGE of the HCAR3-Gi1 complex bound to compound 6O. **(B)** Representative micrograph of the complex particles. **(C)** Representative 2D averages. **(D)** Workflow for cryo-EM image processing. **(E)** Gold-standard FSC curves of the 3D reconstructions. **(F)** Local resolution map of the complex. **(G)** Angular distribution calculated in cryoSPARC for the final 3D reconstruction of 6O-HCAR3-Gi1 complex. **(H)** Representative density maps and models (Contour level 4.20 rmsd) for TM1–7 and ECL2 of HCAR3 as well as the α helices of Gαi1 (αN and α5).
(TIF)

**S2 Fig. Cryo-EM data processing of HCAR3-Gi1 signaling complex in the PLA-bound form. (A)** Size exclusion chromatography profile and SDS-PAGE of the HCAR3-Gi1 complex bound to PLA. **(B)** Representative micrograph of the complex particles. **(C)** Representative 2D averages. **(D)** Workflow for cryo-EM image processing. **(E)** Gold-standard FSC curves of the 3D reconstructions. **(F)** Local resolution map of the complex. **(G)** Angular distribution calculated in cryo-SPARC for the final 3D reconstruction of PLA-HCAR3-Gi1 complex. **(H)** Representative density maps and models (Contour level 5.60 rmsd) for TM1–7 and ECL2 of HCAR3 as well as the α helices of Gαi1 (αN and α5).
(TIF)

**S3 Fig. Cryo-EM data processing of HCAR3-Gi1 signaling complex in the IBC293-bound form. (A)** Size exclusion chromatography profile and SDS-PAGE of the HCAR3-Gi1 complex bound to IBC293. **(B)** Representative micrograph of the complex particles. **(C)** Representative 2D averages. **(D)** Workflow for cryo-EM image processing. **(E)** Gold-standard FSC curves of the 3D reconstructions. **(F)** Local resolution map of the complex. **(G)** Angular distribution calculated in cryoSPARC for the final 3D reconstruction of IBC293-HCAR3-Gi1 complex. **(H)** Representative density maps and models (Contour level 3.90 rmsd) for TM1–7 and ECL2 of HCAR3 as well as the α helices of Gαi1 (αN and α5).
(TIF)

**S4 Fig. Cryo-EM data processing of HCAR3-Gi1 signaling complex in the acifran-bound form. (A)** Size exclusion chromatography profile and SDS-PAGE of the HCAR3-Gi1 complex bound to acifran. **(B)** Representative micrograph of the complex particles. **(C)** Representative 2D averages. **(D)** Workflow for cryo-EM image processing. **(E)**Gold-standard FSC curves of the 3D reconstructions. **(F)** Local resolution map of the complex. **(G)** Angular distribution calculated in cryoSPARC for the final 3D reconstruction of Acifran-HCAR3-Gi1 complex. **(H)** Representative density maps and models (Contour level 4.70 rmsd) for TM1–7 and ECL2 of HCAR3 as well as the α helices of Gαi1 (αN and α5).
(TIF)

**S5 Fig. Cryo-EM data processing of HCAR2-Gi1 signaling complex in the acifran-bound form. (A)** Size exclusion chromatography profile and SDS-PAGE of the HCAR2-Gi1 complex bound to acifran. **(B)** Representative micrograph of the complex particles. **(C)** Representative 2D averages. **(D)** Workflow for cryo-EM image processing. **(E)** Gold-standard

FSC curves of the 3D reconstructions. **(F)** Local resolution map of the complex. **(G)** Angular distribution calculated in cryoSPARC for the final 3D reconstruction of Acifran-HCAR2-Gi1 complex. **(H)** Representative density maps and models (Contour level 4.30 rmsd) for TM1–7 and ECL2 of HCAR2 as well as the α helices of Gαi1 (αN and α5).
(TIF)

**S6 Fig. Map to model FSC curves. (A)** Map to model FSC curve of 6O-HCAR3 complex. **(B)** Map to model FSC curve of PLA-HCAR3 complex. **(C)** Map to model FSC curve of IBC293-HCAR3 complex. **(D)** Map to model FSC curve of Acifran-HCAR3 complex. **(E)** Map to model FSC curve of Acifran-HCAR2 complex.
(TIF)

**S7 Fig. Properties of ligand binding pocket. (A)** Cross-section of HCAR3 binding pockets in the presence of 6O, PLA, IBC293, and acifran. N-terminal, ECL1, and ECL2 are marked with yellow, purple, and cyan, respectively. **(B)** 2D presentation of agonist chemical structures and interactions with HCAR3 residues. Residues in pink rectangle: polar interactions (hydrogen bonds, salt bridge). Residues in yellow rectangle: π–π interactions. Residues in green rectangle: hydrophobic interactions.
(TIF)

**S8 Fig. Electron density of key residues and ECL2 involved in ligand binding. (A)** Density of ligand-interacting residues and ECL2 in the 6O-HCAR3 complex (Contour level 6O 4.20 rmsd). **(B)** Density of ligand-interacting residues and ECL2 in the PLA-HCAR3 complex (Contour level PLA 5.60 rmsd). **(C)** Density of ligand-interacting residues and ECL2 in the IBC293-HCAR3 complex (Contour level 6O 3.90 rmsd). **(D)** Density of ligand-interacting residues and ECL2 in the acifran-HCAR3 complex (Contour level acifran 4.70 rmsd).
(TIF)

**S9 Fig. Mutagenesis study of ligand-interacting residues. (A)** Additional hydrophobic interaction with L30[1.35], L34[1.39], F277[7.36], and L280[7.39] in 6O-bound HCAR3 structure. **(B)** Effect on Gi-mediated cAMP by single point mutation of several residues that interact with 6O. **(C, D)** Cell surface expression level of wild-type and single point mutant HCAR3 **(C)** and HCAR2 **(D)**. The data are presented as means±SEM. The experiments are performed in triplicates. The underlying data for this figure can be found in S1 Data.
(TIF)

**S10 Fig. Analysis of the polar and hydrophobic interactions in different agonist-bound HCAR3 complexes. (A)** Superposition of 4 HCAR3 complexes aligned based on the receptor regions. The orthosteric binding pockets were highlighted with the dashed circle. **(B)** π–π interactions between the aromatic moiety of ligands and F107[3.32]. **(C)** Surface view of R1 residues from 6O-bound and acifran-bound structures. R1 of acifran-bound structure is obviously looser than R1 of 6O-bound structure. **(D)** Surface view of R2 residues from 6O-bound and acifran-bound structures. R2 of 6O-bound structure formed a groove-like surface. **(E)** IBC293 lacked salt bridge interactions with R111[3.36] while acifran possesses. **(F)** Effect on IBC293-induced HCAR3 activation by single point mutation of R111[3.36]A. **(G)** Surface view of R1 residues from IBC293-bound and acifran-bound structures. The underlying data for this figure can be found in S1 Data.
(TIF)

**S11 Fig. Binding affinity of the wild-type HCAR3 for ligands. (A)** Binding affinity of the wild-type HCAR3 for agonist 6O. **(B)** Binding affinity of the wild-type HCAR3 for agonist PLA. **(C)** Binding affinity of the wild-type HCAR3 for agonist IBC293. **(D)** Binding affinity of the wild-type HCAR3 for agonist acifran. Binding affinity is determined by surface plasmon resonance analysis. The data are presented as means±SEM. The experiments are performed in triplicates. The underlying data for this figure can be found in S1 Data.
(TIF)

**S12 Fig. Molecular mechanism of Acifran recognition and activation of HCAR2 versus HCAR3. (A)** Binding affinity of the wild-type HCAR2 for agonist acifran. **(B)** HCAR2 key residues interacting with acifran and effects on HCAR2 activation by single point mutation of these residues. **(C, D)** Detailed comparison of R1 residues between acifran-HCAR2 and acifran-HCAR3 structures. Benzene group of acifran fit R1 better in HCAR2 than in HCAR3. **(E)** Acifran induced cAMP inhibition of WT HCAR2 and homology-mutated HCAR2, WT HCAR3 and homology-mutated HCAR3. **(F)** Acifran stability evaluation in HCAR2 and HCAR3. RMSD value Raw (light color) and smoothed (10 ns moving average, dark color). Red dashed line is the average RMSD value for three replica trajectories. The underlying data for this figure can be found in S1 Data.
(TIF)

**S13 Fig. Electron density of key residues and ECL2 involved in Acifran binding to HCAR2. (A)** Electron density of polar and hydrophobic residues in HCAR2 that interact with acifran (Contour level 4.30 rmsd). **(B)** Electron density of ECL2 in HCAR2 (Contour level 4.30 rmsd).
(TIF)

**S14 Fig. Pocket difference between HCAR3 and HCAR2. (A)** Surface representation of HCAR3 and HCAR2 orthosteric pocket, as well as superimposed HCAR2 and HCAR3 orthosteric pockets. The structures and agonists are colored differently. Medium aquamarine and light salmon, acifran-HCAR2; light green and pale violet red, 6O-HCAR3. Blue, HCAR2 surface; yellow, HCAR3 surface. **(B–D)** Effects on 6O, IBC293, PLA-induced HCAR3/HCAR2 activation by homology mutation of $83^{2.60}$, $86^{2.63}$, $91^{23.48}$, and $107^{3.32}$. The data are presented as means±SEM. The experiments are performed in triplicates. The underlying data for this figure can be found in S1 Data.
(TIF)

**S15 Fig. Helix displacement of ligand-bound HCAR3 complexes compared to inactive HCAR2 structure. (A)** Comparison of 6O-, PLA-, IBC293-bound HCAR3 structures with inactive HCAR2 structure (PDB:7ZL9). **(B)** Displacements of TM7 upon receptor activation in the structures of 6O-, PLA-, IBC293-bound HCAR3 with inactive HCAR2. **(C)** Analysis of the mechanism of TM7 interacted with α5 helix of Gi.
(TIF)

**S16 Fig. Activation mechanism of HCAR3. (A, B)** Shift of TM5 and TM6 in agonist-bound HCAR3 compared to inactive HCAR2 (PDB: 7ZL9). The structures are colored differently: gray, inactive-HCAR2; pale violet red, 6O-HCAR3; slate blue, PLA-HCAR3; medium sea green, IBC293-HCAR3; dodger blue, acifran-HCAR3. **(C–F)** The key $C^{6.47}F^{6.48}xP^{6.50}$ (CWxP motif in common GPCRs) **(C)**, $P^{5.50}I^{3.40}F^{6.44}$ **(D)**, $D^{7.49}P^{7.50}xxY^{7.53}$ (NPxxY motif in common GPCRs) **(E)**, and $D^{3.49}R^{3.50}Y^{3.51}$ **(F)** motifs displayed conformational rearrangement in activated HCAR2 and HCAR3.
(TIF)

**S17 Fig. Electron density of motif residues. (A)** Density of $C^{6.47}F^{6.48}xP^{6.50}$(CWxP motif in common GPCRs) motif residues. **(B)** Density of $P^{5.50}I^{3.40}F^{6.44}$ motif residues. **(C)** Density of $D^{7.49}P^{7.50}xxY^{7.53}$ (NPxxY motif in common GPCRs) motif residues. **(D)** Density of $D^{3.49}R^{3.50}Y^{3.51}$ motif residues. Pale violet red, 6O-HCAR3; slate blue, PLA-HCAR3; medium sea green, IBC293-HCAR3 (Contour level 6O 4.20 rmsd, PLA 5.60 rmsd, IBC293, 3.90 rmsd).
(TIF)

**S18 Fig. Mechanism diagram of ligand recognition and selectivity of HCAR3 and HCAR2.** This study primarily elucidates the mechanisms by which the agonists 6O, PLA, and IBC293 bind to HCAR3. 6O is the ligand with the highest binding affinity for HCAR3. 6O, PLA, and IBC293 exhibit strong binding to HCAR3 but weaker binding to HCAR2. The key mechanism lies in the fact that the binding pocket of HCAR3 is larger than that of HCAR2. F/L107$^{3.32}$, V/L83$^{2.60}$, Y/N86$^{2.63}$, and S/W91$^{3.48}$ are the most critical residues determining the ligand selectivity between HCAR2 and HCAR3.

(TIF)

**S19 Fig. Electron density map comparison between reported structure and structure from this article. (A)** Comparison of electron density for ligand acifran and key residues related to acifran selectivity from Suzuki and colleagues (PDB: 8IHJ) and us. **(B, C)** Comparison of electron density for ECL1 and ECL2 hairpin regions from Suzuki and colleagues and us.
(TIF)

**S1 Table. Cryo-EM data collection, refinement, and validation statistics.**
(PDF)

**S2 Table. Parameters setting for MD simulation.**
(PDF)

**S1 Raw Images.** Uncropped Coomassie-stained SDS–PAGE gel used for S1A, S2A, S3A, S4A, and S5A Figs.
(PDF)

**S1 Data.** The raw data for Figs 2C, S9B, S9C, S9D, S10F, S11A, S11B, S11C, S11D, S12A, S12B, S12E, S14B, S14C, and S14D.
(XLS)

## Acknowledgments

We would like to thank the Kobilka Cryo-Electron Microscopy Center, the Chinese University of Hong Kong, Shenzhen for our cryo-electron microscopy. We would like to thank the Warshel Institute for Computational Biology (funding from Shenzhen City and Longgang District) for computational work.

## Author contributions

**Conceptualization:** Fang Ye, Yang Du.

**Data curation:** Fang Ye, Zhiyi Zhang, Binghao Zhang, Xinyu Li.

**Formal analysis:** Fang Ye, Zhiyi Zhang, Binghao Zhang, Xinyu Li, Jiaxi Deng, Qian Miao, Peiruo Ning, Yunlin Chi, Geng Chen, Zhangsong Wu, Qian Wang, Lezhi Xu, Ningjie Gong, Bangning Cheng, Xin Pan.

**Funding acquisition:** Fang Ye, Zhigang Ma, Yang Du.

**Investigation:** Fang Ye, Zhiyi Zhang, Binghao Zhang, Xinyu Li, Jiaxi Deng, Qian Miao, Peiruo Ning, Yunlin Chi, Geng Chen, Zhangsong Wu, Qian Wang, Lezhi Xu, Ningjie Gong, Bangning Cheng.

**Methodology:** Fang Ye, Zhiyi Zhang, Binghao Zhang, Xinyu Li.

**Project administration:** Fang Ye, Binghao Zhang, Xinyu Li, Yang Du.

**Resources:** Fang Ye, Yang Du.

**Software:** Fang Ye, Zhiyi Zhang, Binghao Zhang, Xinyu Li.

**Supervision:** Zhigang Ma, Chungen Qian, Lizhe Zhu, Xin Pan, Yang Du.

**Validation:** Fang Ye, Binghao Zhang, Yang Du.

**Visualization:** Fang Ye, Binghao Zhang, Yang Du.

**Writing – original draft:** Fang Ye, Zhiyi Zhang, Binghao Zhang, Xinyu Li, Xin Pan, Yang Du.

**Writing – review & editing:** Fang Ye, Binghao Zhang, Zhigang Ma, Yang Du.

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
