## [Editor Report · Decision Letter 0]

9 Apr 2025

Dear Yang,

Thank you for submitting your manuscript entitled "Structural insights into the ligand selectivity of the human hydroxycarboxylic acid receptor HCAR3 and HCAR2" for consideration as a Research Article by PLOS Biology. Please accept my apologies for the delay in getting back to you as we consulted with an academic editor about your submission.

Your manuscript has now been evaluated by the PLOS Biology editorial staff, as well as by an academic editor with relevant expertise, and I am writing to let you know that we would like to send your submission out for external peer review.

IMPORTANT: After discussions with the editorial team, we think your manuscript would be a better fit as a Short Report at the journal (https://journals.plos.org/plosbiology/s/what-we-publish#loc-short-reports). Upon resubmission (details below), I would be grateful if you could please tick 'Short Report' as the article type in the dropdown menu.

Before we can send your manuscript to reviewers, we need you to complete your submission by providing the metadata that is required for full assessment. To this end, please login to Editorial Manager where you will find the paper in the 'Submissions Needing Revisions' folder on your homepage. Please click 'Revise Submission' from the Action Links and complete all additional questions in the submission questionnaire.

Once your full submission is complete, your paper will undergo a series of checks in preparation for peer review. After your manuscript has passed the checks it will be sent out for review. To provide the metadata for your submission, please Login to Editorial Manager (https://www.editorialmanager.com/pbiology) within two working days, i.e. by Apr 11 2025 11:59PM.

Kind regards,

Richard

Richard Hodge, PhD

rhodge@plos.org

PLOS

---

## [Decision Letter · Decision Letter 1]

9 Jun 2025

Dear Dr Du,

Thank you for your patience while your manuscript "Structural insights into the ligand selectivity of the human hydroxycarboxylic acid receptor HCAR3 and HCAR2" was peer-reviewed at PLOS Biology. Please accept my sincere apologies for the delays that you have experienced during the peer review process. Your manuscript has now been evaluated by the PLOS Biology editors, an Academic Editor with relevant expertise, and by three independent reviewers.

In light of the reviews, which you will find at the end of this email, we would like to invite you to revise the work to thoroughly address the reviewers' reports.

As you will see, the reviewers are generally supportive of your work and note that the manuscript provides important insights into the ligand selectivity of HCAR2 and HCAR3. However, the reviewers raise overlapping concerns about the novelty of the study, given that HCAR2/3 cryo-EM structures bound to acifran have been previously reported. After discussions with the academic editor, we agree with Reviewer #2 and we think that the manuscript should be reframed to focus on the HCAR3 structures bound to other agonists as the main advance. In addition, Reviewer’s #2 and #3 raise concerns that the quality of the cryo-EM maps do not allow for unambiguous assignment of ligand poses. A successful revision should address this by providing MD simulation data to provide additional support for the proposed binding modes.

Given the extent of revision needed, we cannot make a decision about publication until we have seen the revised manuscript and your response to the reviewers' comments. Your revised manuscript is likely to be sent for further evaluation by all or a subset of the reviewers.

**IMPORTANT - SUBMITTING YOUR REVISION**

*Re-submission Checklist*

*Published Peer Review*

*PLOS Data Policy*

*Blot and Gel Data Policy*

Best regards,

Richard

Richard Hodge, PhD

rhodge@plos.org

REVIEWS:

Reviewer #1: In the article "Structural insights into the ligand selectivity of the human hydroxycarboxylic acid receptor HCAR3 and HCAR2" Ye F, et al solved the cryoEM structures of HCAR3 and HCAR2 bound to a variety of drugs and analyzed the ligand selectivity between two receptors. Their results indicate 1) HCAR3 contains a larger orthosteric binding pocket to accommodate bigger agonists whereas HCAR2 cannot, 2) the flexibility of ECL1 and the presence of F107 in HCAR3 confer its adaptability to ligands with different chemical properties. The conclusion of this study is supported by functional assays including cAMP ELISA and SPR in addition to MD simulations. Overall, this is a carefully designed and succinctly presented report explaining the mechanisms of action for various HCAR3 selective agonists.

Below list a few minor suggestions.

1) In the introduction, there is no mentioning of previous cryoEM structures of active HCAR2 and HCAR3. A clear description of previous studies and their shortcomings lays solid foundation for unanswered questions that only strengthen the significance of current study. I would like to see the authors acknowledge peers' work.

2) Some figures are misleading or difficult to visualize. For example, Fig.2A presents four ligands at different view angles that causes trouble for this reviewer to compare. Fig.3A and Fig. 4B are so messy that this reviewer still could not distinguish features apart.

3) In the results section, I find "Activation mechanism of HCAR3" is unrelated to the main points of this study. It is also not mentioned in the abstract, introduction, or discussion section of this manuscript. Furthermore, both the active HCAR2 and HCAR3 structures have been reported previously, so this section's novelty is low. Unless further explained, I suggest move this section to supplementary materials so that the article is more focused on ligand selectivity.

Reviewer #2 (Xuan Zhang, signs review): The manuscript "Structural insights into the ligand selectivity of the human hydroxycarboxylic acid receptor HCAR3 and HCAR2" by Ye et al., presents five cryo-EM structures of HCAR3-Gi complexes with agonists compound 6O, PLA, IBC293 and acifran, as well as HCAR2-Gi complex with agonist acifran. The authors clarified the specific binding pocket by examining the binding modes of different ligands in HCAR3 and discussed the subtype selectivity of ligands between HCAR3 and HCAR2. This work provides extensive insights into the ligand selectivity and activation mechanism of HCAR3 and HCAR2, potentially advancing drug development targeting HCAR3.

There are several concerns which need to be addressed:

1. The structures of human HCAR3-Gi and HCAR2-Gi bound with the non-selective agonist acifran closely resemble those reported in a 2023 Nature Communications paper (https://doi.org/10.1038/s41467-023-41650-7). Therefore, this section should be significantly shortened. In contrast, the structures of human HCAR3-Gi bound with selective agonists 6O, PLA, and IBC293 represent the novelty and significance of this study.

2. The structures were determined by cryo-EM at overall resolutions of 2.7-3.3 Å. According to the local resolution estimations shown in the Supplementary Figures, the resolution in the extracellular region is quite low. Therefore, the cryo-EM densities for the critical residues and extracellular loops (ECLs) involved in ligand binding should be shown.

3. PLA and IBC293 are relatively small, while the corresponding cryo-EM density are very large. Figure 1C and 1D also suggest that PLA and IBC293 may be modeled in the opposite direction in the HCAR3. How did the authors validate the correct orientation of these ligands?

4. Homology mutation experiments and MD simulations indicated that acifran possessed higher stability in HCAR2 pocket than in HCAR3's. However, the authors should also demonstrate the significance of IBC293, 6O, and PLA to HCAR3 relative to HCAR2 through similar homology mutation experiments and MD simulations.

5. The angle distribution maps should be included in Supplementary Figures to provide a comprehensive assessment of the final map quality. Additionally, Figure 3A should be clearly labeled for better clarity.

Reviewer #3: The manuscript by Ye et al describes the structures of two GPCRs, HCAR2 and HCAR3, in complexes with heterotrimeric Gi protein, bound to several agonists: HCAR3-compound 6O, -D-phenyllactate, -IBC293, -acifran, HCAR2-acifran. The authors show that 6O occupies the R1 and R2 positions of the orthosteric binding pocket in HCAR3, explaining the high affinity of the drug. The results also suggest the principles of ligand selectivity between HCAR2 and HCAR3, explained the pi-pi interaction with residue F107 in HCAR3, pocket complementarity (size and specific residues in place).

Overall I think this is a very well crafted manuscript, with high quality illustrations and a consistent story. While some of the structures reported here have previously been published (e.g. HCAR2-acifran, Suzuki et al), the authors argue that their study provides a more complete picture with a better resolved structure. This is a reasonable argument in my opinion.

My main concern is that the quality of the maps does not allow unambiguous placement of the ligands into the corresponding densities with absolute certainty. This is evident from figure 1B-F. For pretty much each of the compounds, one can propose alternative poses. For 6O, which has a tripod-like structure, potentially at least 3 poses might be possible. For PLA and IBC293, one could imagine rotating the molecules by 180 degrees along two axes, potentially leading to 4 poses per ligand. For acifran in both HCAR2 and HCAR3, two poses might be possible based on the map alone. The authors performed the MD simulations for their chosen poses, but they have not attempted to do the same for the alternative poses of the drugs - the energy of interaction / RMSDs of the compound atoms during the MD simulation would have been robust metrics for selecting the "right" pose. It is of course possible that some logic was applied in placing each of the compounds in a specific orientation / pose based on the properties of the binding sites. If so, this should be clarified and ideally still supplemented with MD of the "wrong" poses for good measure.

Since the whole story relies completely on the interpretations of the chosen ligand poses, I think the validity of the chosen ligand poses is a fundamental issue that needs to be resolved. In case the alternative poses are actually possible and are equally stable as the ones chosen by the authors - this should be clearly described. In that case the manuscript will have to be re-written to reflect the potential ambiguity in the interpretations.

Minor points:

Map to model FSC curves are missing for all structures. For completeness, please provide those.

Fig 4B right panel and Supplementary Fig 11A right panel are somewhat confusing, with multiple elements placed on top of each other - I am sure this figure can be simplified. Maybe it is easier to digest this material without the complex alignment of the surfaces - just the ligands. Or perhaps showing the structures side by side would simplify the presentation.

In the same figure panels, and also in Fig 3A the ripple effect on the surfaces of the helices is visually slightly distracting. This is really a minor point and a matter of taste.

A summary figure that outlines the key findings in a sketch would be very useful.

---

## [Decision Letter · Decision Letter 2]

9 Oct 2025

Dear Yang,

Thank you for your patience while we considered your revised manuscript "Structural insights into the ligand selectivity of the human hydroxycarboxylic acid receptor HCAR3 and HCAR2" for publication as a Short Report at PLOS Biology. This revised version of your manuscript has been evaluated by the PLOS Biology editors, the Academic Editor and two of the the original reviewers.

Based on the reviews, I am pleased to say that we are likely to accept this manuscript for publication, provided you satisfactorily address the following data and other policy-related requests that I have provided below (A-F):

(A) Please note that we considering your manuscript as a Short Report, which has a maximum of 4 main figures. Therefore, we ask that the number of main figures is reduced at this stage by either combining main figures or moving one of the figures to the Supplementary (Figure 5?).

(B) We routinely suggest changes to titles to ensure maximum accessibility for a broad, non-specialist readership. In this case, we would suggest a minor edit to the title, as follows. Please ensure you change both the manuscript file and the online submission system, as they need to match for final acceptance:

“Structures of G-protein coupled receptor HCAR3 in complex with selective agonists reveal the basis for ligand recognition and selectivity”

(C) Thank you for providing the structural data in the PDB and EMDB databases. However, we note that the data is currently on hold for release. We ask that you please make the structures publicly available at this stage before publication.

(D) Please also ensure that each of the relevant figure legends in your manuscript include information on *WHERE THE UNDERLYING DATA CAN BE FOUND*, and ensure your supplemental data file/s has a legend.

(E) Per journal policy, if you have generated any custom code during the course of this investigation, please make it available without restrictions. Please ensure that the code is sufficiently well documented and reusable, and that your Data Statement in the Editorial Manager submission system accurately describes where your code can be found.

(F) Please note that per journal policy, the model system/species studied should be clearly stated in the abstract of your manuscript.

We expect to receive your revised manuscript within two weeks.

*Published Peer Review History*

*Press*

Best regards,

Richard

Richard Hodge, PhD

rhodge@plos.org

Reviewer remarks:

Reviewer #2 (Xuan Zhang, signs review): The authors have satisfactorily addressed all of my comments. I can now support its publication in PLOS Biology.

Reviewer #3: The authors have addressed all of my concerns in a satisfactory manner.

---

## [Editor Report · Decision Letter 3]

21 Oct 2025

Dear Yang,

On behalf of my colleagues and the Academic Editor, Yan Zhang, I am pleased to say that we can accept your manuscript for publication, provided you address any remaining formatting and reporting issues. These will be detailed in an email you should receive within 2-3 business days from our colleagues in the journal operations team; no action is required from you until then. Please note that we will not be able to formally accept your manuscript and schedule it for publication until you have completed any requested changes.

In addition, I would be grateful if you could please make the structural data deposited in the PDB and EMDB databases publicly available during the production process. Please note that we will be unable to publish your manuscript until this data has been released according to our Data Availability Policy.

PRESS

Best wishes,

Richard 

Richard Hodge, PhD

rhodge@plos.org

PLOS
